# The Application of Quantitative Metabolomics for the Taxonomic Differentiation of Birds

**DOI:** 10.3390/biology11071089

**Published:** 2022-07-21

**Authors:** Ekaterina A. Zelentsova, Lyudmila V. Yanshole, Yuri P. Tsentalovich, Kirill A. Sharshov, Vadim V. Yanshole

**Affiliations:** 1Laboratory of Proteomics and Metabolomics, International Tomography Center SB RAS, Institutskaya 3a, 630090 Novosibirsk, Russia; zelentsova@tomo.nsc.ru (E.A.Z.); lucy@tomo.nsc.ru (L.V.Y.); yura@tomo.nsc.ru (Y.P.T.); 2Laboratory of Molecular Epidemiology and Biodiversity of Viruses, Federal Research Center of Fundamental and Translational Medicine, Timakova Str. 2, 630117 Novosibirsk, Russia; sharshov@yandex.ru

**Keywords:** quantitative metabolomics, phylogeny, hierarchical clustering analysis, NMR spectroscopy, eye lens, birds

## Abstract

**Simple Summary:**

Modern evolutionary biology offers a wide variety of methods to explore the evolution of species and to describe their relationships. The methods of DNA/RNA sequence analysis have been developing for decades and have become increasingly popular and reasonably reliable. Nevertheless, final phylogenetic trees for many taxa are still under debate because both classical and genomics-based approaches have their own limitations for phylogenetic tree reconstruction. Here, we propose the use of younger ‘omics’ methods, namely quantitative metabolomics, to aid the phylogeny reconstruction of vertebrates. We show that metabolomics-based hierarchical clustering analysis trees match, although not perfectly, to the genomics-based trees.

**Abstract:**

In the current pilot study, we propose the use of quantitative metabolomics to reconstruct the phylogeny of vertebrates, namely birds. We determined the concentrations of the 67 most abundant metabolites in the eye lenses of the following 14 species from 6 orders of the class Aves (Birds): the Black kite (*Milvus migrans*), Eurasian magpie (*Pica pica*), Northern raven (*Corvus corax*), Eurasian coot (*Fulica atra*), Godlewski’s bunting (*Emberiza godlewskii*), Great crested grebe (*Podiceps cristatus*), Great tit (*Parus major*), Hawfinch (*Coccothraustes coccothraustes*), Hooded crow (*Corvus cornix*), House sparrow (*Passer domesticus*), Rock dove (*Columba livia*), Rook (*Corvus frugilegus*), Short-eared owl (*Asio flammeus*) and Ural owl (*Strix uralensis*). Further analysis shows that the statistical approaches generally used in metabolomics can be applied for differentiation between species, and the most fruitful results were obtained with hierarchical clustering analysis (HCA). We observed the grouping of conspecific samples independently of the sampling place and date. The HCA tree structure supports the key role of genomics in the formation of the lens metabolome, but it also indicates the influence of the species lifestyle. A combination of genomics-based and metabolomics-based phylogeny could potentially resolve arising issues and yield a more reliable tree of life.

## 1. Introduction

To explore the evolution of species and to describe their relationships, traditional taxonomy, for a long time (since Linnaeus and Darwin in the 18–19th centuries), was based on systemically analyzing the traits of species with respect to their morphology, physiology and behavior [1]. Currently, evolutionary biologists obtain most of their knowledge of the phylogeny of organisms through the cladistic analysis of morphological characters [2], which has been reinforced at the end of the 20th century by a wide variety of methods based on the analysis of nucleotide sequences [3]. Advances in genomics and the development of DNA/RNA sequence databases as well as computational bioinformatics algorithms have made it possible to develop taxonomy science and to apply the genomics methodology to molecular phylogenetic reconstruction [3]. Moreover, such approaches allow for the construction of relevant models for taxon diversification time, which is crucial for the phylogenetic reconstruction of the tree of life.

The morphological traits of animals are closely connected with biomolecules—nucleic acids, proteins and metabolites—therefore, biomolecules have become useful additional indicators of phylogenetic relationships between species [3,4]. Molecular data are easily converted into a numerical form, and thus they can be stored in databases and consequently are amenable to further mathematical and statistical analysis. Significant points that should be considered when analyzing the genomic data include the following: (1) a relevant gene needs to be chosen for the phylogenetic tree construction; (2) the chosen gene should be sequenced for all species under investigation; (3) in order to make a relevant tree, it is often not sufficient to use a single gene; and (4) tree construction should be performed with a relatively large group of species and should include genetically distant species to yield reliable results. Genomics-based phylogeny has already been applied to resolve trees for many species [5]. Methods of analysis are continuously improving, and with newer phylogenetic methods and high throughput sequencing, these points start to be resolved.

Today, many scientists are in favor of a more integrative taxonomy [6,7,8], and they are in search of different sets of characters to describe phylogeny. Such an integrative approach can potentially solve arising limitations and can allow scientists to reconstruct a more reliable tree of life. The development of younger ‘-*omics*’ methods, such as metabolomics, can contribute to the more reliable construction of phylogenetic trees. Several recent papers have proposed to use the metabolomic approach for the classification of species. This approach has primarily been applied to plant species and subspecies [9,10,11,12,13,14,15] (the approach has been known in phytochemistry since the 1960s as ‘chemotaxonomy’) and to a lesser extent to fungi species [16,17], animal microbiomes [18] and human microbiomes [19]. Among these papers, some authors used an integrative approach—a combination of metabolomics with genome sequence analyses [10,11,12,16,18,19]. In all cases, metabolomics gave fruitful results, allowing for the differentiation of samples. Especially valuable outcomes were found for lower taxonomic levels [13]. No noticeable papers on the metabolomic phylogeny reconstruction of other kingdoms, especially of vertebrates, were found. It is worth noting that the metabolomic data in these papers are semi-quantitative or qualitative, so the re-use of these data meets serious limitations, which are discussed later. The application of a genomics-based approach to the reconstruction of the tree of life for birds [20,21,22,23,24,25,26,27] follows the general trend in taxonomy. Bird taxonomy is considered to be intrinsically hard to resolve [1], as the Avian tree structure was determined at an early stage in the evolutionary history of birds because of the rigorous limitations placed upon flying vertebrates [28]. Thus, the Avian tree likely contains a hard polytomy at the base of Neoaves [29]. One of the latest trees was constructed in a recent extensive paper by Kuhl H et al. [27] by the application of a novel 3′-UTR-based phylotranscriptomics method for 429 bird species from 379 genera.

The phylogenetic dendrogram is a hypothesis that is often debated for many taxa, because both classical and molecular approaches have their own limitations for relevant interpretation and phylogenetic reconstruction [1,27,30,31]. For example, Kuhl H et al. [27] reported that, in the Neoaves, the Mirandornithes are the sister taxon of all other taxa, which contradicts all previous molecular trees [21,22,24,25,26]. Therefore, various taxonomic and phylogenetic approaches should be used with care. One of the modern trends in taxonomy suggests the use of a combination of several molecular methods for phylogenetic tree reconstruction, and conflicts between hypotheses should be conciliated by the analysis of different sets of characters or even by analysis with a combination of sets of characters.

In the current paper, we propose the use of quantitative metabolomics for the taxonomic differentiation of vertebrates. For the successful application of metabolomics for resolving phylogeny, suitable organs and tissues of organisms should be selected for sampling and further analysis beforehand. Eye lenses seems to be a promising tissue for this purpose. The lens is the transparent tissue of an individual, which is anatomically isolated from other tissues [32]. To be transparent, the lens tissue consists of densely packed fiber cells without nuclei and organelles, and the metabolic activity inside the lens is minimal. For this reason, the protection of the lens tissue from oxidative, osmotic and other stresses is almost entirely provided by metabolites [33,34]. The metabolite exchange between blood and the lens proceeds via aqueous humor, and the composition of the majority of metabolites in the lens reflects the smoothed in time metabolomic profile of the whole body [32]. At the same time, some important compounds, including antioxidants, osmolytes and energy metabolites, are synthesized in the metabolically active lens epithelial monolayer [32], and the composition of these metabolites indicates the lens-specific requirements related to the animal genetic features, lifestyle and feeding behavior [33,35,36].

This paper is a pilot interdisciplinary study directed at the possibility of using quantitative metabolomics and corresponding conventional unsupervised statistical approaches (principal component analysis and hierarchical clustering analysis) for the differentiation between species of the Animalia kingdom in comparison with their phylogenetic relationships (i.e., phylometabolomics [37]). To this end, we determined the concentrations of the most abundant metabolites in the eye lenses of 14 species from 6 orders of the class Aves (Birds): Accipitriformes, Columbiformes, Gruiformes, Passeriformes, Podicipediformes and Strigiformes. Our preliminary results have shown that the topology of metabolomics-based dendrograms resemble a phylogenetic tree. It was interesting to compare the topology of metabolomics-based and genomics-based dendrograms. For better representation, the current study includes both genetically close and distant bird species. In particular, eight species, including the Eurasian magpie (*Pica pica*), Northern raven (*Corvus corax*), Hooded crow (*Corvus cornix*), Rook (*Corvus frugilegus*), Godlewski’s bunting (*Emberiza godlewskii*), Great tit (*Parus major*), Hawfinch (*Coccothraustes coccothraustes*) and House sparrow (*Passer domesticus*), belong to the Passeriformes order. Four out of these eight species (*P. pica*, *C. corax*, *C. cornix* and *C. frugilegus*) correspond to the Corvidae family, and the other four correspond to the Passerides infraorder. Other birds, including the Black kite (*Milvus migrans*), Eurasian coot (*Fulica atra*), Great crested grebe (*Podiceps cristatus*), Rock dove (*Columba livia*), Short-eared owl (*Asio flammeus*) and Ural owl (*Strix uralensis*), represent non-passerine orders. All species under analysis are rather widespread in Siberia (Russia), and the samples were possible to obtain.

At the molecular level, the adaptations of species are imprinted in all layers of the functional organization of life: the genome, transcriptome, proteome and metabolome. Among these layers, the genome is the most conservative, whereas the metabolome is the most susceptible to the variability of many external and internal factors. Consequently, the composition and the concentrations of metabolites in the lens, or the lens metabolome, are determined by the combination of conservative (e.g., genomic) and variable (e.g., lifestyle and feeding) factors and thus is expected to vary to some extent between species. It is interesting to establish the magnitude of such variations and to evaluate the contributions of conservative and variable factors into the formation of the total metabolome of the lens.

The major goals of the current study are: (1) to identify the most abundant metabolites in the lenses of 14 bird species from 6 orders and to establish their concentrations; (2) to analyze the metabolomic composition of bird lenses; (3) to assess the applicability of the generally used statistical metabolomic approaches to the phylogenetic differentiation between species and to compare the genomics- and metabolomics-based trees; and (4) to evaluate the influence of conservative and variable factors on the total metabolome of the lens.

## 2. Materials and Methods

### 2.1. Chemicals

Chloroform and methanol HPLC grade were purchased from Panreac (Barcelona, Spain). D_2_O 99.9% was purchased from Armar Chemicals (Döttingen, Switzerland). Acetonitrile HPLC grade of quality 0 was purchased from Cryochrom (St. Petersburg, Russia). All other chemicals were of the highest purity, usually HPLC or LC-MS grade, and were purchased from Sigma-Aldrich (Burlington, MA, USA). H_2_O was deionized using an Ultra Clear UV plus TM water system (SG water, Barsbüttel, Germany) to the quality of 18.2 MOhm.

### 2.2. Lens Sample Collection and Species Description

The study was conducted in accordance with the ARVO Statement for the Use of Animals in Ophthalmic and Vision Research and the European Union Directive 2010/63/EU on the protection of animals used for scientific purposes, and with the ethical approval from the International Tomography Center SB RAS (ECITC-2017-02).

The bird species were collected from three sources (Table 1): (1) during the hunting season with a license from the regional Ministries of Ecology and Natural Resources (Altay Republic; Tyva Republic; Novosibirsk Region, Russia) as part of the annual collection of biological material under the program for the study of infectious diseases of wild animals with the approval of the Biomedical Ethics Committee of FRC FTM, Novosibirsk, Russia (Protocols No. 2013-23 and 2021-10); (2) provided by the Center for the Rehabilitation of Wild Animals (CRWA, Novosibirsk, Russia) after the humane euthanasia of mortally wounded birds; and (3) sampled with special permission to catch for scientific purposes from the Committee for the Protection of the World’s Wild Animals of the Republic of Altay, Russia (#5, 21 August 2018).

The species under analysis are common in Siberia, and the samples were relatively accessible. All species were adults of wild origin and were obtained within the 2017–2020 time period. Lenses were extracted clean from the bodies, placed into separate cryotubes, frozen in liquid nitrogen and kept at −70 °C until the analysis within the 2017–2021 time period. Depending on the size of a lens, each sample contained one to three lenses from different individuals (Table 1).

### 2.3. Sample Preparation

The lens sample preparation was performed by the procedure that has been described in detail [36,39,40]. We analyzed either one lens per sample or pooled together two or three lenses from different individuals. Each sample was weighed prior to homogenization. The typical sample and lens weights are given in Table 1. Lenses were placed in glass vials and homogenized with a TissueRuptor II homogenizer (Qiagen, Netherlands) in 1600 µL of cold (−20 °C) MeOH, and then 800 µL of water and 1600 µL of cold chloroform were added. The mixture was shaken well in a shaker for 20 min and was left at −20 °C for 30 min. Then, the mixture was centrifuged at 16,100× *g*, +4 °C for 30 min, yielding two immiscible liquid layers separated by a protein layer. The upper aqueous layer (MeOH-H_2_O) was collected, divided into two parts for NMR (2/3) and LC–MS (1/3) analyses and lyophilized.

### 2.4. NMR Measurements

The extracts for the NMR measurements were re-dissolved in 600 μL of D_2_O containing 2 × 10^−5^ M sodium 4,4-dimethyl-4-silapentane-1-sulfonic acid (DSS) as an internal standard and 50 mM deuterated phosphate buffer to maintain a pH of 7.2. The ^1^H NMR measurements were carried out at the Center of Collective Use “Mass spectrometric investigations” SB RAS with the use of an NMR spectrometer AVANCE III HD 700 MHz (Bruker BioSpin, Ettlingen, Germany) equipped with a 16.44 Tesla Ascend cryomagnet, as described in [39]. The concentrations of the metabolites in the samples were determined by the peak area integration relative to the internal standard DSS.

### 2.5. LC–MS Measurements

In this work, LC–MS data were used only for the identification of several unknown compounds and for the confirmation of the data obtained by the NMR method. The extracts for the LC–MS analysis were re-dissolved in 100 μL of 10 mM ammonium formate and 0.1% formic acid solution in H_2_O. The LC separation was performed with an UltiMate 3000RS chromatograph (Dionex, Bremen, Germany) using hydrophilic interaction liquid chromatography (HILIC) on a TSKgel Amide-80 HR (Tosoh Bioscience, Griesheim, Germany) column (4.6 × 250 mm, 5 μm). The MS detection was performed with an ESI-Q-TOF high-resolution hybrid mass spectrometer maXis 4G (Bruker Daltonics, Bremen, Germany), as described earlier [34,40].

### 2.6. Metabolomic Data Analysis

The statistical treatment of the quantitative metabolomics data—principal component analysis (PCA), hierarchical cluster analysis (HCA) and heatmap construction—was performed at the MetaboAnalyst 5.0 web-platform (www.metaboanalyst.ca (accessed on 10 January 2022) [41]) with the data either non-scaled, auto-scaled (mean-centered and divided by the standard deviation of each metabolite concentration) or Pareto-scaled (mean-centered and divided by the square root of the standard deviation of each metabolite concentration). Auto-scaled and Pareto-scaled methods were used to normalize the contributions of the metabolites in further analyses.

For the construction of dendrograms based on hierarchical clustering, several parameters related to the HCA data treatment need to be chosen: the method of the data normalization (e.g., non-scaled, auto-scaled and Pareto-scaled), the type of the distance\similarity measure (e.g., Euclidian, Spearman and Pearson), and the clustering algorithm (e.g., Ward, Average, Complete and Single). The HCA parameter values listed above are available as built-in options at the widely-used-in-metabolomics web platform MetaboAnalyst v5.0, where the current analysis was performed. It should be noted that other tree-building software or the *Python/R* statistical packages allow for choosing a wider variety of HCA parameter values (e.g., Manhattan distance, Bray–Curtiss dissimilarity, WPGMA clustering, etc.). However, the current pilot work is aimed at assessing the general applicability of the quantitative metabolomics-based HCA method for phylogeny. Thereby, we decided not to go deep into developing the correct HCA parameters, and thus we used only the built-in MetaboAnalyst ones. Moreover, we did not restrict possible combinations of values. The general guideline for choosing the appropriate HCA parameters is to use ones that maximize the distance between samples in different classes and to minimize that within each class.

### 2.7. Phylogenetic Tree Reconstructions for Birds from the Literature

In a preliminary analysis of the metabolomics-based dendrograms, we observed similarity in the topology with the published phylogenetic dendrograms based on different approaches. To test this hypothesis and to compare the metabolomics-based trees and the “classical” genomics- or transcriptomics-based phylogeny, we manually reconstructed two phylogenetic trees from the literature to fit only the 14 bird species under study. The backbone parts of a tree from the class (Aves) to the orders (Columbiformes, Podicipediformes, Gruiformes, Accipitriformes, Strigiformes, and Passeriformes) were based on two recently proposed modern bird phylogenies, either on work of Jarvis ED et al. [24] or on the newer work of Kuhl H et al. [27]. The vast order *Passeriformes* includes 8 out of 14 species under investigation, and a more detailed tree for this order was adapted from the work of Oliveros CH et al. [42]. To this purpose, we removed the clades not containing the species under study and connected the remaining branches keeping the tree structure. The time scale and the time of divergence were discarded, and the resulting tree was rather schematic for displaying the phylogenetic relationships.

## 3. Results

### 3.1. The Identification and Quantification of Metabolites

The identification of metabolites was performed according to their NMR spectra that are available in the literature, in databases (HMDB, METLIN, BMRB and SpectraBase) and in our in-house NMR library [33,35,39]. In some cases, when NMR signal assignment was not obvious, we spiked the lens extract with commercial standard compounds and validated the identification of metabolites. For the identification of unknown signals, we also used the fractioning of the metabolomic extract by HPLC followed by the MS and NMR analysis of each fraction, as described in [40]. Nevertheless, several signals in NMR spectra remained unidentified. These signals, together with the identified metabolites, are included in Table 2. They are annotated as S109, S112, S120, D121, D139, T727 and S823 (Appendix A), where the letter indicates the multiplicity of the signal (S for singlet, D for doublet and T for triplet), and the numerals show the chemical shift of the signal (for example, S109 corresponds to the singlet at 1.09 ppm).

The concentrations of the metabolites in the lenses (in nmol/g) were calculated by the integration of the NMR signals relative to the internal standard DSS followed by normalization to the tissue wet weight. Typically, 60–80 compounds were identified for every species; however, the NMR signals from some compounds were either too weak or strongly overlapped by other signals, which made the quantification of these compounds unreliable. For that reason, the final set of metabolites studied in this work was restricted to 67 identified compounds and 7 unknowns. For every species, except *A. flammeus* and *S. uralensis*, the measurements were performed for 3–12 individuals (Table 1, Appendix A), and the results were averaged. For the rare species *A. flammeus* and *S. uralensis*, the lenses from only one individual were analyzed. The obtained data expressed as the mean ± standard deviation (SD) are collected in Table 2. The relative abundances of 10 major (in average) metabolites in the lenses of 14 species under study are shown in Figure 1. In Appendix A, the most abundant metabolites from Table 2 are sorted in descending order and are highlighted in each species separately to facilitate further analysis: a red color indicates a metabolite concentration above 10 µmol/g, a yellow color indicates a concentration between 3 and 10 µmol/g and a green color indicates a concentration between 1 and 3 µmol/g.

### 3.2. General Overview of Bird Lens Metabolomes

There were two principal metabolites in the bird lenses with an average concentration above 20 µmol/g: *myo*-inositol and taurine. In the lens of *A. flammeus*, the concentration of lactate was also slightly above 20 µmol/g. Both *myo*-inositol and taurine are well-known osmolytes [34,35,43,44], protecting lens fiber cells from osmotic stress. In all species except *P. cristatus*, *S. uralensis* and *A. flammeus*, both osmolytes share the function of osmotic protection and have very high abundance above 10 µmol/g (Table 2, Appendix A). In the lenses of *F. atra*, *C. corax*, *P. pica*, *M. migrans*, *C. cornix*, *C. frugilegus* and *C. livia*, *myo*-inositol has a higher abundance than taurine (up to 2.5 times); in *P. cristatus*, *myo*-inositol prevails tenfold over taurine. In all four Passerides (*C. coccothraustes*, *E. godlewskii*, *P. domesticus* and *P. major*), taurine has a higher abundance than *myo*-inositol (up to 1.9 times), and in both Strigidae (*S. uralensis* and *A. flammeus*), taurine prevails 5–10 times over *myo*-inositol.

Eight further metabolites had average concentrations below 10 and above 3 µmol/g, namely, lactate, glutamine, acetate, glutathione, alanine, ergothioneine, serine and ATP. These compounds can be ascribed to the major metabolites of bird lenses (Appendix A). High levels of these metabolites indicate their important roles in cellular processes. In living cells, including metabolically inert lens fiber cells, they are involved in various functioning: osmotic protection (*myo*-inositol, taurine, glutamine and serine); antioxidant protection (glutathione and ergothioneine); and cellular energy generation (ATP, acetate, glutamine and alanine).

There were six metabolites with concentrations above 1 µmol/g in more than one species (Appendix A): glutamate, creatine, pyroglutamate, glucose, phosphocholine and valine. Concentrations above 1 µmol/g in only one species were found for: ADP and glycine (*P. domesticus*); proline and betaine (*E. godlewskii*); methionine and anserine (*F. atra*); leucine and D139 (*P. cristatus*); UDP (*A. flammeus*); and NADH (*M. migrans*).

The composition of the most abundant metabolites vary from species to species, and the most pronounced differences in comparison with other species are observed for both birds from the Strigidae family, *A. flammeus* and *S. uralensis*, and for *P. cristatus* (Figure 1). It should be also noted that the levels of major metabolites in all passerine species are rather similar (Figure 1, Appendix A), indicating the importance of the genetic factor in the formation of the tissue metabolome.

### 3.3. Principal Component Analysis (PCA)

PCA is the method of choice in metabolomics to take a glance at the obtained results and to display general similarities and differences in the data. Figure 2a shows a PCA scores plot based on auto-scaled data for all bird species studied in this work.

One can see that the conspecific samples are grouped together in the plot, although they are often not clearly separated from other species along the first two principal components (PC1 and PC2). It is worth noting that, independently of the sampling place and date (e.g., *C. corax*, Table 1), the grouping of conspecific samples is observed. Moreover, such a grouping is observed in the PCA plots independently of the data scaling (Appendix A, for non-scaled and Pareto-scaled PCA). The most distant species in the plot (Figure 2a) are *A. flammeus*, *S. uralensis* and *P. cristatus*. All birds from the Passeriformes order form a cluster at the upper left part of the plot, and inside this cluster, the groups of species belonging to the Corvides and the Passerides infraorders are visibly separated. One of the main advantages of PCA is that the analysis is unsupervised. However, it works well for a limited number of groups of samples. When the number of groups increases, the discrimination between groups (if it exists) may start to vanish in the PC1 vs. PC2 plot. In this case, the search for the discrimination between groups can include the use of additional dimensions with subsequent principal components (PC3, PC4, etc.) or reductions in the number of groups by removing distant groups. In the present work, we performed PCA for 67 samples from 14 groups (Figure 2, Appendix A). The grouping of conspecific samples is rather good, but the discrimination between groups in the PC1 vs. PC2 plot is unreliable. For better discrimination of unresolved species, we removed phylogenetically distant species and left the Passeriformes order only (39 samples, 8 groups); the resulting PCA scores plot is presented in Figure 2b. Both infraorders, the Corvides and the Passerides, in Figure 2b, are well-separated. Inside the Passerides infraorder, a separation between species and the grouping of the same species are clearly visible; *P. domesticus* samples are now the most distant from other Passerides. However, no clear separation is observed for the Corvides.

### 3.4. Hierarchical Clustering Analysis (HCA) and Heatmaps

We performed the analysis for all 36 possible combinations of available HCA parameter values (Appendix A). The quality of the obtained dendrograms was monitored according to the following criteria: conspecific samples should be in one cluster and should not mix with other species; the Passerides and the Corvides samples should form larger clusters and link with each other to form the Passeriformes cluster; and *P. cristatus* and *C. livia* should be distant from other clusters.

Fairly good quality of dendrograms was obtained for most combinations of the HCA parameters. As an example, the HCA tree plotted with the default set of built-in parameters, with auto-scaled data plotted with the Euclidean distance similarity measure and using Ward’s linkage clustering algorithm (AEW), is demonstrated in Figure 3. The quality of the tree is rather high. Conspecific samples are clustered together (except *P. pica* and *C. cornix*), and Passerides and the Corvides samples form a larger Passeriformes cluster. Similar to the PCA analysis, the grouping of the conspecific samples is observed independently of other factors, such as age, sex or sampling place and date (e.g., *C. corax*, Table 1). Similarly, good clustering was obtained with non-scaled and Pareto-scaled data plotted with the Spearman’s rank correlation similarity measure and with the use of a single-linkage clustering algorithm (NSS and PSS, Appendix A). Nevertheless, many non-scaled and Pareto-scaled HCA trees show rather poor clustering of conspecific samples and further positioning of larger clusters due to the data scaling.

Data normalization (scaling) is often required to equalize the contributions of high- and low-abundant metabolites into analysis; if one uses non-scaled data, the position of the species in the dendrogram is determined by the concentration variation for the several most abundant metabolites only. The exception from this rule is the use of non-scaled data with nonparametric statistics (e.g., Spearman), since the latter discards the information on the concentrations. On the other hand, taking into account the information on metabolite abundance in some specific form can potentially be helpful for phylogeny. For example, for PSS clustering (Appendix A), the position of the *P. cristatus* species is the most correct compared to the other HCA trees, and the samples are the most distant from other clusters.

Heatmaps add a second visual dimension to the HCA analysis. Appendix A shows a heatmap chart constructed for the AEW clustering (Figure 3). The advantage of this type of result presentation in comparison with the HCA dendrogram is that the additional information is visualized; the metabolites participating in the sample clustering and dendrogram construction are clearly seen in the heatmap. Appendix A shows that approximately twenty compounds (leucine, NADPH, creatine, etc.; upper-left corner) in *P. cristatus* differ significantly from other species. For *C. coccothraustes*, two unknowns (S112 and S120) are flaring. For both Strigidae species, 10–11 metabolites (NAA, valine, phenylalanine, etc.; lower-left corner in Appendix A) stand out. The *F. atra* species has an elevated amount of asparagine, anserine and carnosine.

### 3.5. Genomics- and Transcriptomics-Based Schematic Phylogenetic Tree Construction from the Literature

To test the assumption of similarity in the topology of metabolomics-based HCA dendrograms and the phylogenetic dendrogram of the studied species, we manually constructed two schematic literature-based phylogenetic dendrograms. The first tree, reconstructed from phylogenomic research by Jarvis ED et al. [24] and Oliveros CH et al. [42], is presented in Figure 4a. The second tree is based on newer extensive phylotranscriptomics research from Kuhl H et al. [27] (Figure 4b). The main difference between Kuhl H et al. and Jarvis ED et al. trees, related to our 14 bird species, is that, in the Kuhl H et al. tree, there is no node between the Columbiformes and the Podicipediformes, which corresponds to the Columbea clade in the Jarvis ED et al. tree. Moreover, the Columbiformes order in the Kuhl H et al. tree forms a cluster with the Gruiformes, forming the Basal landbirds clade. These differences are marked with arrows in Figure 4.

## 4. Discussion

It can be safely said that birds rely deeply on their eyesight for living. The optical system of the eye is evolutionarily adapted for the specific needs of a bird [45]. The adaptation factors include, but are not limited to, the habitat, lifespan, feeding behavior, hunting behavior and diurnal or nocturnal lifestyle. It can be expected that adaptations strongly influence the morphology and composition of the optical apparatus, particularly the lens. A significant part of these adaptations is secured at the genomic level, but the feeding behavior and lifestyle of certain bird species may also have a noticeable influence, especially at the metabolomic level.

In modern metabolomics, three general types of data can be distinguished: qualitative, semi-quantitative and quantitative data. Qualitative data either determine the presence of metabolites in a tissue or answer the question of whether the relative content of metabolites is higher or lower between the two groups of samples. The semi-quantitative approach is aimed at the quantitative (numeric) comparison of the relative content of metabolites in different groups of samples. Most often, semi-quantitative data are obtained with the use of electrospray ionization-based mass spectrometry (ESI-MS). The ESI-MS signal intensity is not directly proportional to the metabolite concentration; metabolites with a low concentration but a high ionization ability can give more intense signals than metabolites with a high concentration but a low ionization ability. In addition, matrix effects and ionization suppression can lead to the biased interpretation of data [33,46].

The quantitative approach yields absolute concentrations of metabolites in a tissue (e.g., in moles per gram). It demands much more effort and time than qualitative or semi-quantitative methods, but the data obtained have a long-term value. In addition, the results of such experiments are available for future ‘eternal’ re-use, such as for data mining, new interpretations, the addition of new samples or groups, new comparisons, etc. [47]. That cannot be achieved with ESI-MS semi-quantitative data; it is almost impossible to add new samples to the previous experiments due to the limitations of ESI-MS instruments. For quantitative metabolomics, the method of proton nuclear magnetic resonance spectroscopy (^1^H NMR) is often used. Peak areas of the metabolite signals in NMR spectra are directly proportional to the compound concentrations, which makes the determination of the metabolite levels in a tissue easy and straightforward. The results of the present work, obtained with the use of NMR-based quantitative metabolomics, demonstrate that the data obtained can be used not only for the comparison of the biological features of different species, but also for animal classification and for studying the evolution of biochemical processes in living nature.

We have found that the composition and concentrations of the most abundant metabolites vary from species to species, but their levels in genetically close species are very similar, which indicates the importance of the genetic factor in the formation of the tissue metabolome. The eye lenses of all birds contain very high concentrations of two metabolites, *myo*-inositol and taurine, which share the function of lens osmotic protection [33,34,43,44]. The ratio between their concentrations also strongly depends on the species phylogeny; genetically close species have a rather similar ratio (Table 1 and Figure 1, Appendix A). An especially pronounced shift in this ratio is observed for both of the Strigidae species (taurine abundance is much higher). Earlier in our lab, we found that the lenses of another nocturnal animal, the Rat (*R. norvegicus*), contain high concentrations of taurine, which prevails almost tenfold over *myo*-inositol (13.2 and 1.8 µmol/g correspondingly) [33]. It has been also recently suggested [48] that, besides osmotic protection, *myo*-inositol in the eye lens may act as a chaperone, protecting the lens proteins from the aggregation caused by post-translational modifications during a lifespan. Most likely, low levels of *myo*-inositol in owl lenses should be attributed to their nocturnal lifestyle, so the light-induced protein modifications for these species are less dangerous.

A number of quantified metabolites in bird lenses, on average, have rather high concentrations (*myo*-inositol, taurine, lactate, glutamine, acetate, glutathione, alanine, ergothioneine, serine, ATP, glutamate, creatine, pyroglutamate, glucose, phosphocholine, valine, ADP, glycine, proline, betaine, methionine, anserine, leucine, UDP and NADH), which emphasizes their significance in important biochemical processes that are genetically adapted to the lifestyle necessities of the species. These high-abundant metabolites are involved in osmotic protection, antioxidant protection and cellular energy generation (glycolysis, the TCA cycle and the urea cycle). It should be noted that a metabolite usually shares several functions and participates in several pathways; thereby, the metabolite role is not limited to the given functions and pathways.

The obtained results show that evolutionary history is a major factor determining the metabolomic composition of a lens. The genetic factor prevails over other factors in sample positioning in the HCA dendrogram: the difference in age, sex, place of catching, etc., for conspecific species influence positioning less than interspecific differences. The following observations also support the predominance of the genetic factor: (A) genetically close bird species have similar concentrations of the most abundant metabolites (Figure 1 and Table 2, Appendix A); (B) in PCA plots, the samples are positioned according to their genetic relationship (Figure 2, Appendix A); and (C) in HCA dendrograms (Figure 3, Appendix A), the species under study form trees very similar to the classic Avian trees (Figure 4). The last argument needs more detailed consideration. Although no examined HCA tree ideally fits into the phylogenetic trees (Appendix A), the general phylogenetic dependencies are well-reproduced. Figure 5 shows the comparison of the simplified phylogenetic trees and the dendrograms AEW, NSS and PSS constructed according to the metabolomic data.

The following features can often be observed for the obtained dendrograms (Figure 5):Conspecific samples are positioned together and form clusters separated from other species (Figure 3, Appendix A).All four Passerides (*C. coccothraustes*, *E. godlewskii*, *P. domesticus*, *P. major*) are positioned together in a separate larger cluster.All four Corvides (*P. pica*, *C. corax*, *C. cornix*, *C. frugilegus*) are positioned together in a single cluster.The Passerides and the Corvides samples are positioned in two connected branches, forming a larger cluster (Passeriformes).Birds of prey (*M. migrans*, *A. flammeus*, and *S. uralensis*), either from the Afroaves clade [24] or from the Higher landbirds clade [27], are positioned nearby, and their samples are not mixed with other species.*C. livia* and *P. cristatus* are distant from other clusters.

The following differences between the HCA and phylogenetic trees (shown by arrows in Figure 5) should be mentioned:A node between *C. livia* and *P. cristatus* from the Columbea clade in the Jarvis ED et al. tree [24] does not exist in any HCA dendrogram (dashed green lilac arrow).*F. atra* in the HCA dendrogram is positioned close to the Corvides infraorder, most likely indicating the influence of lifestyle on the metabolomic composition of the *F. atra* eye lens (dark violet arrow); no node between *F. atra* and *C. livia* (the Basal landbirds clade from Kuhl H et al. tree [27]) was found in any HCA dendrogram.Although *C. frugilegus* and *P. pica* are well clustered with the Corvides, they have rather incorrect phylogenetic distances within the Corvides. *C. frugilegus* should be closer to the other species from the Corvus genus (*C. corax* and *C. cornix*), and *P. pica* should be the sister taxon to all Corvus. The samples of *P. pica*, *C. corax* and *C. cornix* often mix together, without the formation of separate clusters for each species (dashed magenta, mint and aquamarine arrows).Similarly, the incorrect positioning is observed for *P. major*; it should be more distant from the other species of the Passerida parvorder (light green arrow).

These observations support the key role of genomics in the formation of the lens metabolome, but they also indicate the influence of lifestyle. In particular, the *F. atra*, *P. Pica* and *P. major* discrepancies in the positioning in the HCA trees most likely originate from lifestyles and feedings. Moreover, our metabolomics data analysis supports the tree structure proposed by Kuhl H et al. better than that by Jarvis ED et al. due to the absence of the Columbea-like clade.

In the current pilot study, we assessed only built-in parameters from the MetaboAnalyst web-platform for the phylogenetic dendrogram construction. Most likely, choosing other types of metrics or clustering algorithms may yield better results. The further development of methods for metabolomics-based tree construction can most likely demand adaptations of other distance-matrix or non-distance-matrix-based methods that are now widely used in phylogenetics, e.g., the maximum likelihood method or parsimony analysis. In addition, the perfect tree should be stable, and bootstrapping-like methods are also required for the adaptation.

## 5. Conclusions

In the current paper, we applied methods of quantitative metabolomics and corresponding statistical approaches for the differentiation between 14 species from 6 orders of the class Aves (Birds) and for further phylometabolomic tree construction. We determined the concentrations of the most abundant metabolites in the eye lenses of the species and deposited the corresponding raw NMR spectra and the metabolomic analysis into our Animal Metabolite Database repository (https://amdb.online (accessed on 10 January 2022)). The most fruitful results were obtained with the hierarchical clustering analysis. The topology of the obtained dendrograms is very similar to the topology of genetics-based trees, and general phylogenetic dependencies are well-reproduced, although none of the HCA dendrograms ideally fit to the genetics-based trees. The HCA dendrogram structure supports the key role of genomics in the formation of the lens metabolome, but it also indicates the influence of lifestyle. Very likely, most discrepancies in the HCA dendrograms as compared to genomics-based phylogenetic trees originate from different lifestyles and feedings.

Perhaps, the addition of metabolomic data for a larger number of bird species into the analysis, as well as the development of HCA methods, can produce a clearer correlation and similarity of the phylogeny of birds with eye lens metabolomics-based hierarchical clustering. A combination of phylogenetic and phylometabolomic analyses can potentially solve issues in the reconstruction of bird phylogeny and can yield a more reliable tree of life. Current methods can be applied to differentiate other species of vertebrates, and with several adaptations to other species of the animal kingdom and even of other kingdoms, the only limitation is that the tissue under analysis is conservative and comparable between all species under analysis.

## Figures and Tables

**Figure 1 biology-11-01089-f001:**
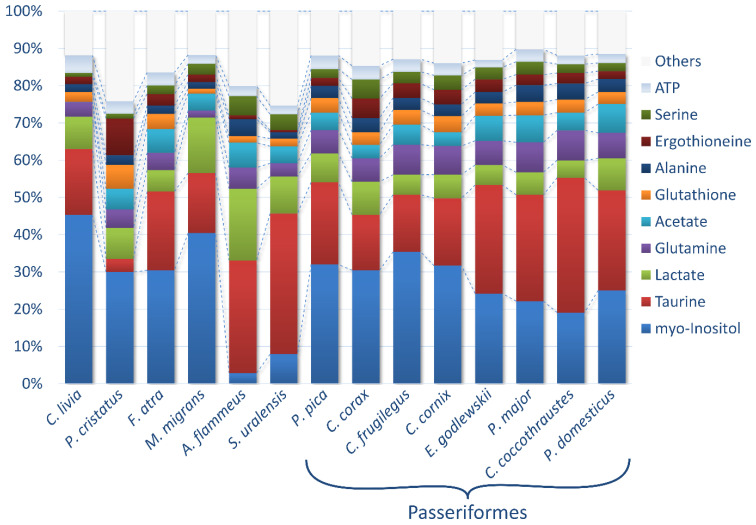
Relative abundances of 10 major metabolites. The concentrations of metabolites in the lenses of the 14 species under study were averaged and sorted in descending order. The remaining metabolites were summed as “Others”.

**Figure 2 biology-11-01089-f002:**
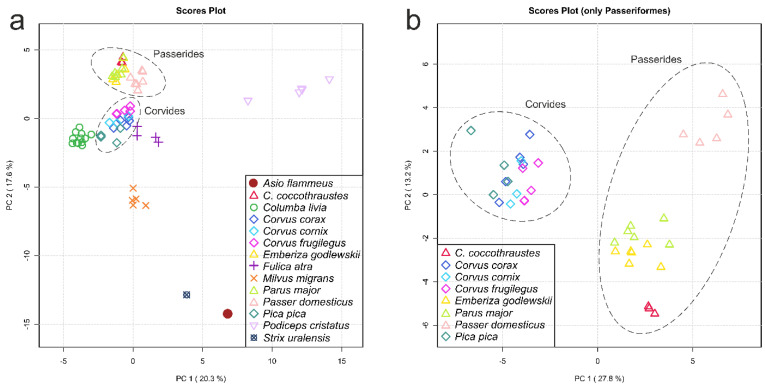
Scores plots of principal component analysis (PCA): (**a**) PCA based on the metabolomic profiles of eye lenses for 14 bird species; (**b**) the Passeriformes only. The data are auto-scaled. Dashed ovals indicate the Passerides and the Corvides infraorders. Variances explained by the first (PC1) and second (PC2) principal components are indicated on the axes of the scores plots.

**Figure 3 biology-11-01089-f003:**
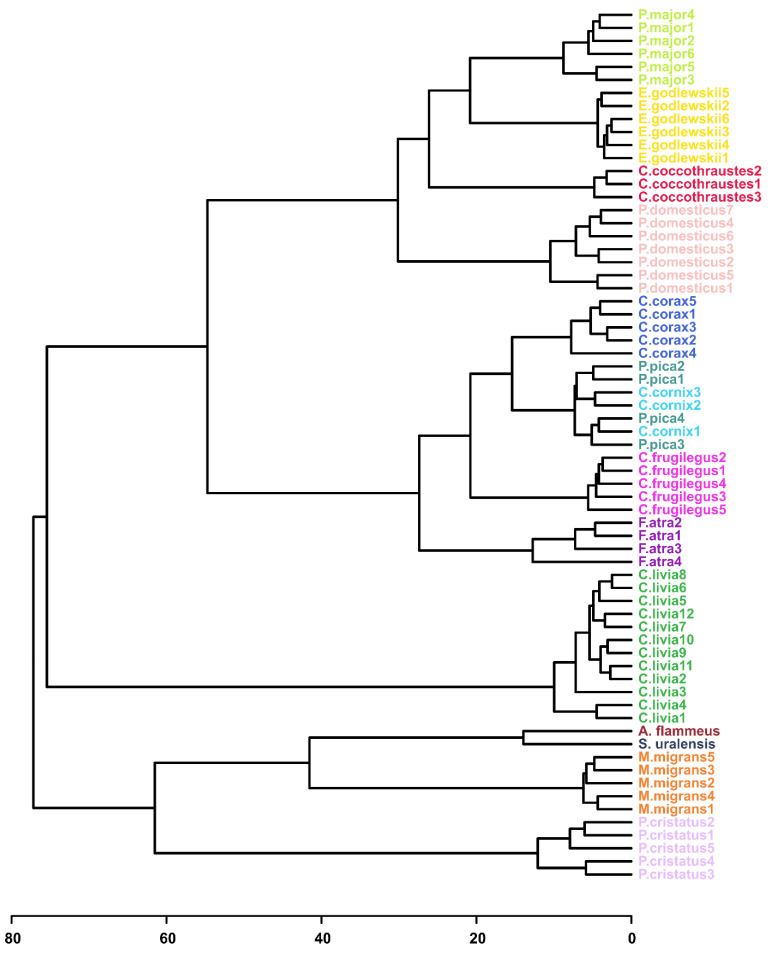
The HCA result shown as a dendrogram. The data are auto-scaled, the distance similarity measure is Euclidean and the clustering algorithm is Ward’s linkage (AEW).

**Figure 4 biology-11-01089-f004:**
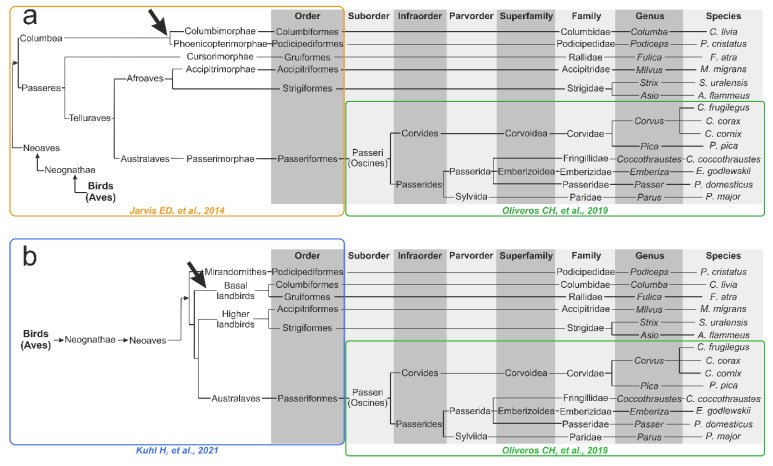
Schematic representation of phylogenetic dendrograms, manually reconstructed from the literature: (**a**) Backbone part of a tree from the class to the orders based on the work of Jarvis ED et al. [24] (yellow rectangle); (**b**) Backbone part of a tree from the class to the orders based on the work of Kuhl H et al. [27] (blue rectangle). The Passeriformes part was adapted from the work of Oliveros CH et al. [42] (green rectangles). The arrow indicates the main difference between the 2 trees related to the 14 bird species under study.

**Figure 5 biology-11-01089-f005:**
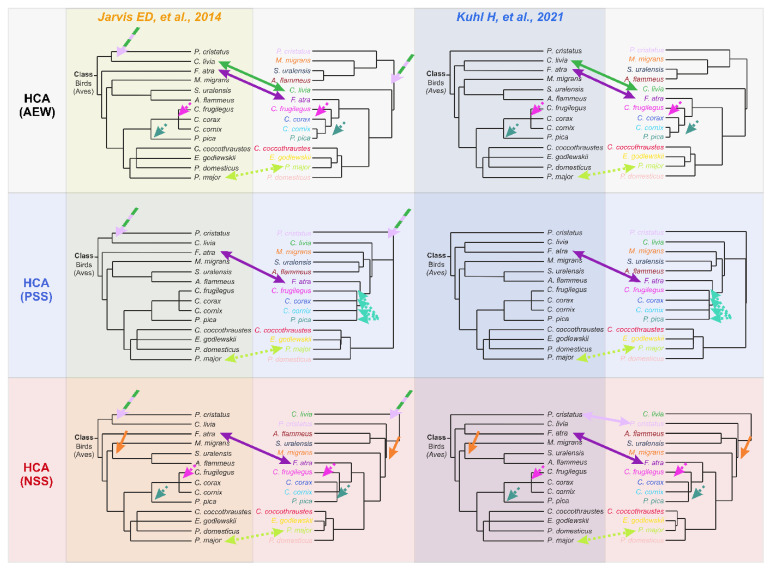
Comparison of metabolomics-based and “classical” dendrograms. Three HCA dendrograms (AEW, PSS and NSS) and two “classical” dendrograms (the phylogenomic-based tree from Jarvis ED et al. [24] and the phylotranscriptomic-based tree from Kuhl H et al. [27]) were analyzed. The dashed green lilac arrow indicates the absence of Columbea-like clade in HCA dendrograms; green and dark violet arrows indicate the significant mispositioning of *C. livia* and *F. atra*, correspondingly; dashed magenta, mint and light green arrows indicate the slight mispositioning of *C. frugilegus*, *P. pica* and *P. major*, correspondingly; dashed aquamarine arrows indicate the mixing of the Corvidae samples in the PSS dendrogram; the orange arrow indicates the absence of the the Afroaves/Higher landbirds-like clade in the NSS dendrogram.

**Table 1 biology-11-01089-t001:** Species and sample descriptions.

Species *	Date and Place of Catching	Typical Lens or Sample Weight, mg	N
Black kite (*Milvus migrans*)	July 2019, Tyva Republic	90–150	5 individuals
Eurasian magpie (*Pica pica*)	December 2018–January 2019, Altay Republic	90–125	4 individuals
Northern raven (*Corvus corax*)	December 2018, Altay Republic; July 2019, Tyva Republic; December 2019, Novosibirsk Region	130–240	5 individuals
Eurasian coot (*Fulica atra*)	April 2019, Novosibirsk Region; May 2019, Tyva Republic	50–90	4 individuals
Godlewski’s bunting (*Emberiza godlewskii*)	January 2019, Altay Republic	60–70 per sample; 20–25 per lens	18 individuals, 6 samples
Great crested grebe (*Podiceps cristatus*)	May 2019, Tyva Republic	70–80	5 individuals
Great tit (*Parus major*)	December 2018, Altay Republic	40–90	12 individuals, 6 samples
Hawfinch (*Coccothraustes coccothraustes*)	December 2018, Altay Republic	70–110 per sample; 30–40 per lens	6 individuals, 3 samples
Hooded crow (*Corvus cornix*)	January 2019, Altay Republic; April 2019, Novosibirsk Region	110–200	3 individuals
House sparrow (*Passer domesticus*)	November 2018, Novosibirsk Region	35–65 per sample; 20–30 per lens	14 individuals, 7 samples
Rock dove (*Columba livia*)	September 2017, Novosibirsk Region	25–55	12 individuals
Rook (*Corvus frugilegus*)	April 2019, Novosibirsk Region	60–105	5 individuals
Short-eared owl (*Asio flammeus*)	May 2019, CRWA, Novosibirsk	156	1 individual
Ural owl (*Strix uralensis*)	May 2019, CRWA, Novosibirsk	164	1 individual

*—English and Latin names are provided according to the IOC World Bird List [38].

**Table 2 biology-11-01089-t002:** Mean concentrations of metabolites in the lenses of 14 bird species. Values are given in nmoles per gram of the tissue wet weight (nmol/g, mean ± SD).

Species	Black Kite	Common Magpie	Common Raven	Eurasian Coot	Godlewski’s Bunting	Great Crested Grebe	Great Tit	Hawfinch	Hooded Crow	House Sparrow	Rock Dove	Rook	Short-Eared Owl ^1^	Ural Owl ^1^
Species Latin	*M. migrans*	*P. pica*	*C. corax*	*F. atra*	*E. godlewskii*	*P. cristatus*	*P. major*	*C. coccothraustes*	*C. cornix*	*P. domesticus*	*C. livia*	*C. frugilegus*	*A. flammeus*	*S. uralensis*
2,3-Butanediol	91 ± 48	56 ± 50	31 ± 35	21 ± 17	19 ± 9	0	11 ± 7	11 ± 9	10 ± 4	130 ± 30	78 ± 34	0	0	0
2-Ketoisovalerate	0	0	0	0	0	0	0	0	0	0	0	0	21	9
2-OH-3-Me-but ^2,3^	18 ± 3	0	0	0	0	0	0	0	0	0	0	0	31	18
2-OH-but	28 ± 12	18 ± 6	16 ± 5	14 ± 4	0	0	0	0	16 ± 3	0	0	0	40	13
3-Me-His ^2^	0	0	0	0	0	100 ± 30	0	0	0	0	0	0	0	0
3-OH-but	96 ± 54	300 ± 130	210 ± 100	170 ± 40	250 ± 50	170 ± 50	370 ± 140	440 ± 40	340 ± 280	240 ± 60	91 ± 62	210 ± 180	480	480
3-OH-isovalerate	0	34 ± 8	22 ± 7	66 ± 15	48 ± 10	51 ± 12	51 ± 24	10 ± 4	24 ± 7	44 ± 22	190 ± 60	47 ± 12	45	21
Acetate	4300 ± 1000	4300 ± 800	3000 ± 800	6000 ± 900	8100 ± 1200	5100 ± 600	9700 ± 2700	6200 ± 1800	3100 ± 1400	10,000 ± 3000	89 ± 27	4900 ± 1000	7200	3100
Acetylcarnitine	60 ± 22	0	0	0	0	0	0	0	0	0	0	0	19	22
ADP	560 ± 80	430 ± 40	380 ± 80	630 ± 120	960 ± 170	720 ± 50	1000 ± 200	740 ± 90	300 ± 100	1100 ± 200	470 ± 110	560 ± 150	290	470
Alanine	1700 ± 400	3000 ± 300	3200 ± 400	2100 ± 300	3600 ± 600	2400 ± 1100	6000 ± 800	5700 ± 100	2600 ± 1200	4700 ± 700	1700 ± 600	2900 ± 400	4900	1200
*alpha*-Aminobut	63 ± 11	56 ± 14	90 ± 26	130 ± 110	72 ± 20	220 ± 90	97 ± 45	91 ± 6	84 ± 12	120 ± 20	20 ± 6	53 ± 11	190	90
*alpha*-OH-isobut ^2^	0	11 ± 4	11 ± 8	14 ± 14	0	0	0	0	12 ± 7	0	0	0	60	61
AMP	60 ± 37	28 ± 9	11 ± 9	31 ± 15	12 ± 8	51 ± 7	60 ± 36	10 ± 5	10 ± 2	280 ± 210	48 ± 47	39 ± 21	0	230
Anserine	120 ± 50	220 ± 160	290 ± 90	2400 ± 500	260 ± 20	0	73 ± 15	200 ± 30	240 ± 40	150 ± 60	0	750 ± 260	28	130
Ascorbate	390 ± 90	210 ± 40	230 ± 50	120 ± 20	440 ± 50	490 ± 70	190 ± 50	260 ± 10	160 ± 40	550 ± 70	200 ± 40	180 ± 20	480	370
Asparagine	170 ± 30	130 ± 70	120 ± 30	200 ± 50	0	37 ± 17	0	0	150 ± 30	0	0	54 ± 26	260	110
Aspartate	0	140 ± 50	210 ± 30	180 ± 40	130 ± 20	220 ± 70	160 ± 10	93 ± 14	220 ± 100	130 ± 30	44 ± 21	240 ± 70	100	89
ATP	2200 ± 300	3300 ± 200	3200 ± 300	3300 ± 200	2400 ± 100	3100 ± 300	4500 ± 1100	3100 ± 600	2700 ± 500	3200 ± 600	3800 ± 500	3100 ± 600	2900	1600
Betaine	590 ± 130	380 ± 60	900 ± 280	210 ± 70	1000 ± 200	230 ± 100	640 ± 130	720 ± 80	470 ± 60	800 ± 160	0	650 ± 230	230	110
Carnitine	43 ± 10	39 ± 7	100 ± 20	51 ± 12	0	52 ± 13	0	0	40 ± 4	0	27 ± 4	57 ± 7	140	110
Carnosine	0	60 ± 32	41 ± 32	550 ± 760	0	0	0	0	60 ± 13	0	0	98 ± 77	120	27
Choline	54 ± 8	11 ± 4	19 ± 14	78 ± 26	13 ± 2	85 ± 15	20 ± 8	32 ± 3	8.7 ± 2.5	34 ± 9	18 ± 7	22 ± 7	41	69
Creatine	580 ± 80	630 ± 100	750 ± 80	1000 ± 200	1100 ± 100	2800 ± 400	730 ± 80	1400 ± 100	780 ± 100	1100 ± 200	810 ± 90	830 ± 110	3600	2300
Ergothioneine	1900 ± 800	1900 ± 700	4500 ± 900	3000 ± 500	4100 ± 600	9100 ± 1400	3800 ± 600	3700 ± 600	3200 ± 200	2900 ± 500	1600 ± 300	3600 ± 500	1000	360
Formate	75 ± 11	240 ± 30	83 ± 33	320 ± 80	130 ± 30	320 ± 60	220 ± 110	180 ± 80	68 ± 12	130 ± 20	84 ± 52	230 ± 50	150	100
Fumarate	13 ± 4	23 ± 9	19 ± 4	36 ± 8	29 ± 6	24 ± 6	32 ± 10	10 ± 4	18 ± 2	24 ± 4	25 ± 7	14 ± 5	27	26
Gl-Ph-Choline	53 ± 25	300 ± 30	390 ± 60	20 ± 15	230 ± 50	240 ± 120	270 ± 40	140 ± 30	400 ± 20	140 ± 30	0	240 ± 40	140	41
Glucose	270 ± 200	1400 ± 300	820 ± 390	1400 ± 200	970 ± 230	500 ± 240	1600 ± 400	1900 ± 200	950 ± 200	450 ± 230	1800 ± 500	800 ± 310	960	0
Glutamate	790 ± 70	1800 ± 300	1900 ± 400	1600 ± 300	3900 ± 400	1000 ± 100	2700 ± 300	3000 ± 100	2100 ± 300	2800 ± 300	2000 ± 300	1900 ± 100	1300	1600
Glutamine	1700 ± 300	5800 ± 1400	5400 ± 800	4400 ± 1400	7800 ± 1600	4600 ± 1700	11,000 ± 1000	11,000 ± 1000	6400 ± 300	9200 ± 1100	3200 ± 500	7400 ± 1800	6100	2400
Glutathione	1200 ± 500	3500 ± 600	2900 ± 300	3900 ± 200	4000 ± 400	6000 ± 1000	4700 ± 300	4600 ± 100	3600 ± 700	4400 ± 1200	2100 ± 400	3600 ± 600	1800	1400
Glycerol	220 ± 80	120 ± 100	99 ± 66	260 ± 300	0	240 ± 80	0	0	71 ± 37	0	0	70 ± 44	0	0
Glycine	580 ± 110	310 ± 130	530 ± 140	440 ± 70	530 ± 120	900 ± 250	550 ± 100	950 ± 110	470 ± 90	1100 ± 100	350 ± 70	540 ± 80	150	560
GSSG	330 ± 120	400 ± 80	280 ± 50	380 ± 90	560 ± 150	350 ± 40	500 ± 190	350 ± 60	260 ± 60	940 ± 530	0	280 ± 40	0	0
GTP	120 ± 40	0	270 ± 30	0	150 ± 20	220 ± 30	260 ± 80	110 ± 40	0	210 ± 50	190 ± 70	270 ± 50	320	200
Histidine	120 ± 30	97 ± 12	90 ± 24	110 ± 50	140 ± 20	500 ± 100	210 ± 40	430 ± 80	140 ± 50	310 ± 50	85 ± 34	100 ± 40	190	98
Hypoxanthine	250 ± 40	94 ± 17	50 ± 27	81 ± 11	51 ± 8	130 ± 10	84 ± 20	53 ± 12	66 ± 15	140 ± 50	180 ± 30	73 ± 15	130	240
Inosinate	26 ± 7	0	0	0	0	0	0	0	0	42 ± 23	15 ± 26	0	11	58
Inosine	89 ± 13	17 ± 14	27 ± 28	24 ± 19	34 ± 3	52 ± 14	42 ± 11	29 ± 1	18 ± 13	66 ± 14	7.7 ± 13.9	44 ± 7	0	0
Isobutyrate	8 ± 4.9	7.3 ± 1	4.6 ± 2.3	15 ± 7	8.7 ± 2.3	9.6 ± 4.3	10 ± 5	7 ± 3.6	8 ± 2.6	16 ± 5	1.5 ± 2.5	4.2 ± 2.8	19	11
Isoleucine	82 ± 20	34 ± 5	34 ± 11	73 ± 15	24 ± 6	78 ± 14	22 ± 5	29 ± 10	42 ± 5	46 ± 9	36 ± 9	40 ± 9	240	320
Lactate	14,000 ± 2000	7100 ± 1100	7600 ± 3300	5500 ± 1100	6400 ± 500	7700 ± 1600	8100 ± 1100	6100 ± 200	5300 ± 700	12,000 ± 2000	7100 ± 700	4900 ± 800	21,000	6700
Leucine	190 ± 40	110 ± 10	140 ± 40	160 ± 30	98 ± 18	2200 ± 400	130 ± 30	100 ± 20	160 ± 30	170 ± 20	85 ± 13	120 ± 20	950	740
Lysine	140 ± 40	67 ± 23	62 ± 26	300 ± 50	0	910 ± 140	0	0	65 ± 30	0	12 ± 30	77 ± 18	160	88
Methionine	230 ± 50	430 ± 170	960 ± 240	1000 ± 200	590 ± 110	350 ± 130	470 ± 100	500 ± 30	550 ± 170	610 ± 80	200 ± 50	380 ± 110	480	170
*myo*-Inositol	38,000 ± 2000	29,000 ± 9000	26,000 ± 2000	29,000 ± 4000	29,000 ± 2000	28,000 ± 3000	29,000 ± 1000	25,000 ± 2000	26,000 ± 0	34,000 ± 2000	37,000 ± 4000	32,000 ± 1000	3000	5400
N,N-DMG ^2^	30 ± 13	0	0	0	0	39 ± 8	0	0	0	0	0	72 ± 13	0	0
NAA	0	0	0	0	0	0	0	0	0	0	0	0	270	500
NAD	500 ± 60	250 ± 50	150 ± 40	120 ± 50	100 ± 20	250 ± 30	130 ± 50	50 ± 9	130 ± 40	190 ± 70	220 ± 40	210 ± 20	320	250
NADH	1200 ± 100	48 ± 17	17 ± 10	86 ± 14	8.3 ± 3.6	560 ± 270	13 ± 7	7.7 ± 3.1	28 ± 13	16 ± 8	2.7 ± 5.3	0	0	9
NADPH ^2^	0	0	0	0	0	300 ± 70	0	0	0	0	0	0	0	0
N-Me-His ^2^	0	0	0	0	0	0	0	0	0	0	0	0	200	80
Ph-Choline	170 ± 50	89 ± 59	550 ± 50	340 ± 30	69 ± 14	110 ± 20	64 ± 21	170 ± 20	96 ± 68	210 ± 20	250 ± 30	200 ± 80	3900	3900
Phenylalanine	180 ± 70	100 ± 50	50 ± 14	74 ± 24	39 ± 4	53 ± 15	24 ± 19	49 ± 10	67 ± 7	44 ± 16	41 ± 16	48 ± 18	510	580
Proline	480 ± 110	320 ± 120	600 ± 120	260 ± 60	1000 ± 200	680 ± 240	480 ± 120	580 ± 110	690 ± 150	670 ± 90	650 ± 350	560 ± 140	620	280
Pyroglutamate	560 ± 60	800 ± 190	920 ± 240	920 ± 180	1700 ± 200	1700 ± 200	1200 ± 200	1900 ± 200	1100 ± 100	1300 ± 200	650 ± 120	840 ± 90	340	170
Pyruvate	6.8 ± 1.3	9.3 ± 1	16 ± 4	12 ± 4	9.2 ± 3.2	11 ± 5	16 ± 6	7 ± 0	11 ± 2	15 ± 3	13 ± 3	6.8 ± 2.9	9	14
Sarcosine ^2^	7 ± 4.4	4 ± 2.8	7.8 ± 1.9	15 ± 5	7.8 ± 2.3	22 ± 19	5.7 ± 4.1	6.7 ± 0.6	7.7 ± 2.1	18 ± 4	7.4 ± 3.9	9.8 ± 3.8	35	30
*scyllo*-Inositol	50 ± 12	34 ± 8	45 ± 11	51 ± 22	70 ± 17	750 ± 250	51 ± 10	39 ± 7	44 ± 9	49 ± 8	55 ± 18	54 ± 13	210	170
Serine	2700 ± 200	2200 ± 100	4400 ± 1500	2200 ± 700	3900 ± 400	1200 ± 300	4600 ± 500	3100 ± 300	3200 ± 600	2900 ± 800	900 ± 240	2800 ± 500	5600	2900
Taurine	15,000 ± 3000	20,000 ± 4000	13,000 ± 1000	20,000 ± 4000	35,000 ± 2000	3200 ± 500	38,000 ± 5000	48,000 ± 3000	15,000 ± 3000	36,000 ± 2000	15,000 ± 2000	14,000 ± 1000	32,000	26,000
Threonine	300 ± 80	300 ± 150	420 ± 120	820 ± 180	440 ± 90	860 ± 30	500 ± 80	370 ± 80	480 ± 110	540 ± 40	260 ± 100	220 ± 70	540	120
Tryptophan	71 ± 38	95 ± 44	31 ± 17	65 ± 12	0	0	0	0	37 ± 6	0	0	0	160	130
Tyrosine	200 ± 50	200 ± 80	200 ± 40	180 ± 60	130 ± 40	130 ± 30	230 ± 70	170 ± 10	210 ± 20	170 ± 40	140 ± 40	170 ± 60	260	97
UDP	330 ± 80	310 ± 60	300 ± 50	410 ± 60	260 ± 60	490 ± 30	340 ± 60	210 ± 20	270 ± 60	310 ± 50	150 ± 60	280 ± 50	1600	590
Valine	210 ± 30	68 ± 13	92 ± 24	200 ± 40	100 ± 20	190 ± 10	110 ± 30	95 ± 3	110 ± 30	170 ± 20	110 ± 20	94 ± 18	1100	1200
S109 ^2^	0	0	0	0	0	0	0	0	0	0	95 ± 23	0	0	0
S112 ^2^	0	0	0	0	0	0	0	81 ± 23	0	0	0	0	0	0
S120 ^2^	0	0	0	0	0	0	0	130 ± 30	0	0	0	0	0	0
D121 ^2^	0	0	0	0	0	950 ± 220	0	0	0	0	0	0	0	0
D139 ^2^	0	0	0	0	0	1100 ± 500	0	0	0	0	0	0	0	0
T727 ^2^	0	0	0	0	0	400 ± 180	0	0	0	0	0	36 ± 6	0	0
S823 ^2^	100 ± 40	0	0	0	0	670 ± 160	0	0	0	0	0	0	0	0

**^1^** Lens from only one individual was analyzed. **^2^** Low metabolite identification confidence, not confirmed by chemical standards. Concentrations of unknowns were estimated assuming that the signals in the aliphatic part of NMR spectra (S109, S112, S120, D121 and D139) correspond to a single CH_3_ group, whereas the aromatic signals (T727 and S823) correspond to a CH group. **^3^** Abbreviations: 2-OH-3-Me-but—2-hydroxy-3-methyl-butyrate; 2-OH-but—2-hydroxy-butyrate; 3-Me-His—3-methylhistidine; 3-OH-but—3-hydroxy-butyrate; 3-OH-isovalerate—3-hydroxy-isovalerate; ADP—adenosine diphosphate; alpha-Aminobut—alpha-aminobutyrate; alpha-OH-isobut—alpha-hydroxy-isobutyrate; AMP—adenosine monophosphate; ATP—adenosine triphosphate; Gl-Ph-Choline—glycerophosphocholine; GSSG—glutathione oxidized; GTP—guanosine triphosphate; N,N-DMG—N,N-dimethylglycine; NAA—N-acetyl-aspartate; NAD—nicotinamide adenine dinucleotide; NADH—nicotinamide adenine dinucleotide reduced; NADPH—nicotinamide adenine dinucleotide phosphate reduced; N-Me-His—N-methylhistidine; Ph-Choline—phosphocholine; UDP—uridine diphosphate. A list with the metabolite full names and ChEBI identifiers is provided in Appendix A.

## Data Availability

Raw NMR spectra, the descriptions of specimens and samples, metabolite concentrations and the preliminary metabolomic analysis are available at our Animal Metabolite Database repository, Experiment ID 145 (https://amdb.online/amdb/experiments/145/). All other data are available from the corresponding author upon request.

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
