# Peer review of "The Application of Quantitative Metabolomics for the Taxonomic Differentiation of Birds"

_biology, 2022, doi:10.3390/biology11071089_

Round 1

Reviewer 1 Report

I have read this MS with much interest, mainly because of my interest in avian taxonomy. Initially, I thought authors would present a new way to be used in taxonomy (stricto sensu), i.e., a new set of characters and techniques for delimiting species – something essential because a good taxonomy is necessary for proposing a hypothesis of phylogenetic relationships, biogeographical reconstruction, and conservation. However, the authors state their work by somewhat disregarding the morphology and repeating that morphology vs. molecules' unfruitful speech. Therefore, I suggest they reorganize their work in light of an "integrative taxonomy," which is more beneficial.    

Other concerns relate to the methods they use to compare their results with the existing hypothesis of phylogenetic relationships. Finally, they must provide readers with at least some rationale, based on pertinent literature, for comparting published phylogenies with their dendrograms, which, for me, is much more phenetics.

I believe authors should restrict their work to what it is: a pilot study that explores the exciting possibilities of using a different set of characters in avian taxonomy, particularly in species limits.

I attached an annotated MS with several comments and questions, which I hope authors find useful in reorganizing their work. 

Author Response

Reviewer 1

I have read this MS with much interest, mainly because of my interest in avian taxonomy. Initially, I thought authors would present a new way to be used in taxonomy (stricto sensu), i.e., a new set of characters and techniques for delimiting species – something essential because a good taxonomy is necessary for proposing a hypothesis of phylogenetic relationships, biogeographical reconstruction, and conservation. However, the authors state their work by somewhat disregarding the morphology and repeating that morphology vs. molecules' unfruitful speech. Therefore, I suggest they reorganize their work in light of an "integrative taxonomy," which is more beneficial.    

Other concerns relate to the methods they use to compare their results with the existing hypothesis of phylogenetic relationships. Finally, they must provide readers with at least some rationale, based on pertinent literature, for comparting published phylogenies with their dendrograms, which, for me, is much more phenetics.

I believe authors should restrict their work to what it is: a pilot study that explores the exciting possibilities of using a different set of characters in avian taxonomy, particularly in species limits.

I attached an annotated MS with several comments and questions, which I hope authors find useful in reorganizing their work. 

Response: Thank you very much for very constructive and helpful analysis of the manuscript! Such expert suggestions, which are written in perfect formulations, have rendered invaluable help. We do understand the criticism about points mentioned above (and marked in Manuscript). Actually, we did not aim to present a new rigorous method for use in bird taxonomy. Moreover, in no case would we want to reduce the enormous importance of morphology for taxonomy, which should take into account all the results of interdisciplinary research to distinguish taxa.

As the reviewer correctly noted, our goal was to describe the new interesting data obtained in the light of integrative taxonomy on the example of a number of bird species. In this regard, we have edited the manuscript as suggested by the reviewer, trying to show the new data obtained in the light of the well-known provisions on the birds’ phylogenetics

Undoubtedly, taxonomy should be based primarily on genetics and phenotypic traits. We have toned down categorical statements concerning molecular approaches in taxonomy.

Comments from Manuscript (PDF-file)

Lines 40-43 The idea authors present here is not correct. Scientists did not simply “observe” morphological or whatever traits of animals to describe their [phylogenetic] relationships during early stages of formal biological systematics. At that time, the scientific paradigm was different that that arose with Wallace and Darwin’s works. Such methods consistently chanced with the cladistic thinking by Henning in the 1660s. The molecular genetics offered a new way to explore sets of characters to propose hypothesis of phylogenetic relationships of organisms but did not constitute any “revolution”. Note that most of our knowledge on the phylogeny of organism, notably vertebrates, were has been built based on cladistic analysis of morphological characters, ahs bas been more recently reinforced by phylogenomic. If you want to present a brief review on the thinking and development of biological systematics, please do I by offering readers a concise view of its development, both technical and philosophical, not by stating that a technique or set of characters are “better” or "more reliable".

Line 54 I really would like to see such alleged “consensus”. What I have seen is that many scientists are in favour of a more “integrative systematics”, i.e., those that use different sets of characters. These approach would be very interest in this paper.

Response: Thank you for the excellent comments! If the reviewer kindly allows, we took liberty to include some wordings from the comments into the manuscript text. We also significantly modified the text according to the suggestions.

Line 59 whats does a "good tree' mean?

Response: we replaced “good tree” with “relevant tree”.

Line 70 Please avoid such a jargon. There will be never a “final tree”, simply because we are dealing with hypothesis of such a process. What we can have at beast is a “hypothesis with more explanatory power”

Response: Absolutely agree, corrected

Line 74 Of course, conflicts among hypothesis should be conciliated by new analysis and or analysis of different sets of characters, or even analysis with a combination of sets of characters. 

Response: Thank you for this suggestion, we added additional useful part of the sentence.

Line 78

That is a good point, particularly if taken from a integrative perspective, not as presented in the previous paragraphs.

Response: Thank you! We modified the general line of the previous paragraphs.

Line 88 I suggest not to use this term. This implies an idea of evolution as a ladder, which is not the case. Please, be precise: use vertebrates or whatever. 

Response: changed to vertebrates

Line 93 what does it exactly mean? Be precise

Response: we meant that suitable organs and tissues of organism should be selected for sampling at the first place for the successful application of metabolomics for resolving taxonomy. Corrected

Lines 106-110 It seems for me, based on this part, that authors are probably confusing systematics with taxonomy. It seems that (But we’ll see through their text) that the technique they are present is very useful for delimiting species. Taxonomy is very important because it is through it that we will propose the terminals for the phylogenetic reconstruction. Furthermore, if such a technique is useful in delimiting , then I would expect comments on species concept, specifically, how the data provided by the technique advocated by authors apply in the context of the debate on species concept. 

Response: We have corrected the inaccurate terms throughout the whole manuscript. As to application of the data obtained to the debate on species concept: The data and results obtained are quite novel, and it is probably too early to find their place in this discussion. Besides, being honest, we are not great experts in systematics, phylogeny and taxonomy, and our contribution in this debate would be rather speculative.

Line 110 I understand this is a pilot study; however, you should explain – at least in the Methods – what you have selected those taxa

Response: In the revised version of the manuscript, the explanation of the species choice is given in Introduction, lines 121-133.

Line 111 not "Class Birds". It is Class Aves, or birds

Response: Corrected.

Table 1

I would suggest you offer a source for the nomenclature, both English and Latin

Response: Added source, corrected names, according to this source.

Lines 212-213 Please, explain better:  why did you do that ? It is not clear for me why you should do such a thing. Please offer a justification too. 

Response: When analyzing metabolomics-based trees we observed its similar topology with described phylogenetic dendrogram based on different approaches. We added this notice.

Line 216

It is not “widely accepted”. The work is one of the several recently proposal of modern bird phylogeny. There are others, e.g., Prun et al. 2015. You can choice the proposal by Jarvis et al, as a premise for analysis and discussion. Just this!

Response: corrected

Line 364 ‘tree’ is a generic term, or… I most frequently used for “dendrograms” resulted from phylogenetic analysis.  You have a dendrogram  form a cluster analysis.

Response: modified according to the comment

Line 396 As I stated above, you should explain better why you did it, and based on which rationale. It seems you merged different hypothesis into one, which is not scientifically sound. Or, if you have any reasonable justification for that, please offer it, based on pertinent literature.  

Furthermore, I did not udenstand what you mean by “test hypothesis that similarity”. It seens you are want to compare your results with “intentionally merged” hypothesis, which is not a test 

Response: We agree. The preliminary analysis has shown that topologies are similar. To test the assumption of similarity of topology in metabolomics-based HCA dendrogram and phylogenetic dendrogram of the studied species, we have manually constructed two schematic literature-based phylogenetic dendrograms. We corrected text according to this.

Lines 415-433

A whole paragraph full of statements but none reference to support them. If, for example, “it can be safely said” something, then you can offer reads with at least a few references. Furthermore, I suggest you star your discussion with a summary of the main  results , then proceed to discussing them

Response: Added reference.

Line 447 Those two paragraphs have little, if any at all, to do with the work, i.e., with the aims of the paper. The contents of these two paraphs does not fit well in a discussion of the use of a set of characters in bid (or any other animal). Please, discuss your results in light of your aims.

Response: The current manuscript represents interdisciplinary study, which discusses not solely the taxonomic differentiation, but also metabolites in the lens, and metabolomics methods. The discussion starts from the metabolomics, continuing with discussion of phylogenetic results.

Line 454 please provide a reference and elaboate

Response: Added reference.

Line 457 A nocturnal animal

Response:  corrected

Line 459 What those all information do have with taxonomy?

Response: The current manuscript represents interdisciplinary study, and this part discusses interesting properties of a particular metabolite.

Line 480 If I understood well, you ae comparing the results of a statistical approach with a “phylogenetic three”. Please, provide at least a rationale for such a comparison.

Response: The preliminary analysis has shown that dendrogram topology resemble phylogeny. We decided to test the assumption.

Line 482 One cannot simply compare a dendrogram with a hypothesis of phylogenetic relationships. Or, if you really want to compare, please offer a justification for such a comparison. Strictly speaking, one could not compare hypothesis based on different methods. Your dendrogram could , for example, be considered as a king of ‘pheneticism’. Please elaborate on why you did such comparison and justify it based on pertinent literature

Response: Actually, we compare dendrograms (1) obtained on the basis of statically processed features, our original results and (2) dendrograms from the literature. Despite the fact that they are schematic and adapted in the form of figures, they are also built on the basis of cladistic analysis, in this case genomic traits.

We believe that without such an initial comparison, there will be no point of view on the perspective of further work on more detailed comparative analysis on the metabolome-based dendrogram and the most relevant genome/phenotype-based dendrogram. This is, as you have noticed, a pilot study with its own assumptions and limitations.

Currently, hypotheses of phylogenetic relationships are also based on phylogenetic dendrograms, these dendrograms are constructed by the same methods and have statistical processing and reliability assessment.

Line 544

I confess I though you would explore the use of such a set of characters in taxonomy, i.e., delimiting taxa (e.g. species), something I did not see. The part of “reconstruct” an alleged phylogeny based on the metabolomics is for me much more close to a new type of pheneticism, than to a different form of phylogenetics.

Response: We believe that is exactly first step of what you mean – we could provide differentiation between 14 species from 6 orders of class Birds (Aves) by methods of quantitative metabolomics and corresponding statistical approaches. And we show that we will see similar picture using construction of phylogenetic dendrogram.

Future studies will allow us to see more clear picture on how we can compare/consider together phenetics and phylogenetic

Reviewer 2 Report

This is an innovative and exploratory study about the phylogenetic signal contained in the metabolites in the eye lens of birds. The study shows that the metabolites indeed contain a phylognetic signal but that they may also reflect the life-style of birds (ecological signal). I think this is a worthwhile study that increases our insight into the evolution of birds. I recommend it for publication in Biology.

My only major issue is that I believe some restructuring of the manuscript is necessary. Specifically, the first two paragraphs of section 3.4 describe what the authors did to get the results, so this belongs in the Methods section. Similarly, paragraph 3.5 belongs in the Methods section. Lines 496 to 526 described new results and should be moved to the Results section. A short summary of those fundings should then be discussed in the Discussion.

Edits and minor comments:

Throughout the paper: change “taxonomic” (trees etc) to “phylogenetic” (trees etc)

Line 14: replace “to date, they became” with “have become”

Line 15: debate (singular)

Line 27: have shown = has shown

Line 29: the hierarchical = hierarchical

Line 47: such taxonomy approach allows = such approaches allow

Line 47: reconstructing = reconstruction

Line 62: delete “various”

Line 64-67: change to “Bird phylogeny is considered to be intrinsically hard to resolve [1], as the Avian tree likely contains a hard polytomy at the base of Neoaves [13].

Change reference 13 to:

Suh, A 2016. The phylogenomic forest of bird trees contains a hard polytomy at the root of Neoaves. Zoologica Scripta 45: 50–62.

Line 75: a prudence = care

Line 87, 92, 93: taxonomy = phylogeny

Line 228: unobvious = not obvious

Figure 1: please make sure Passeriformes symbol also includes Pica pica

Line 275: P. Pica = P. pica

Line 281: Eight further metabolites have average …

Line 316: ... are A. flammeus … (delete “the”)

Line 320: With = When

Line 321: increases

Line 325: delete “Such analysis is not always convenient.”

Line 362: In what way are these “fairly good results”?

Line 365: How do you judge that “The quality of the tree is rather high”?

Line 368: I would point out that in all but one pair of species (Pica pica and Corvus cornix) conspecific samples clustered together

Line 394: increased = elevated

Fig. 4b: change “higher landbirds” to “Afroaves”

Line 415: delete “among all animals”

Line 475: change “genotype is the main factor determining the lens metabolomic composition” to “evolutionary history is a major factor of the lens metabolomic composition

Line 496: delete “that”

Lines 498-499: the clustering of individualsof the same species is not shown in Fig. 5 but instead in Fig. 3.

Line 560: of the birds taxonomy = of bird phylogeny

Author Response

Reviewer 2

This is an innovative and exploratory study about the phylogenetic signal contained in the metabolites in the eye lens of birds. The study shows that the metabolites indeed contain a phylognetic signal but that they may also reflect the life-style of birds (ecological signal). I think this is a worthwhile study that increases our insight into the evolution of birds. I recommend it for publication in Biology.

My only major issue is that I believe some restructuring of the manuscript is necessary. Specifically, the first two paragraphs of section 3.4 describe what the authors did to get the results, so this belongs in the Methods section.

Response: We agree with the Reviewer, and moved this paragraph to Methods.

Similarly, paragraph 3.5 belongs in the Methods section.

Response: Here we don’t agree with the Reviewer, as this is the description of the results, but not the way they have been obtained.

Lines 496 to 526 described new results and should be moved to the Results section. A short summary of those fundings should then be discussed in the Discussion.

Response: Here we don’t agree with the Reviewer, as this part only discusses already obtained results.

Edits and minor comments:

Throughout the paper: change “taxonomic” (trees etc) to “phylogenetic” (trees etc)

Line 14: replace “to date, they became” with “have become”

Line 15: debate (singular)

Line 27: have shown = has shown

Line 29: the hierarchical = hierarchical

Line 47: such taxonomy approach allows = such approaches allow

Line 47: reconstructing = reconstruction

Line 62: delete “various”

Line 64-67: change to “Bird phylogeny is considered to be intrinsically hard to resolve [1], as the Avian tree likely contains a hard polytomy at the base of Neoaves [13].”

Change reference 13 to:

Suh, A 2016. The phylogenomic forest of bird trees contains a hard polytomy at the root of Neoaves. Zoologica Scripta 45: 50–62.

Line 75: a prudence = care

Line 87, 92, 93: taxonomy = phylogeny

Line 228: unobvious = not obvious

Figure 1: please make sure Passeriformes symbol also includes Pica pica

Line 275: P. Pica = P. pica

Line 281: Eight further metabolites have average …

Line 316: ... are A. flammeus … (delete “the”)

Line 320: With = When

Line 321: increases

Line 325: delete “Such analysis is not always convenient.”

Line 362: In what way are these “fairly good results”?

Line 368: I would point out that in all but one pair of species (Pica pica and Corvus cornix) conspecific samples clustered together

Line 394: increased = elevated

Line 415: delete “among all animals”

Line 475: change “genotype is the main factor determining the lens metabolomic composition” to “evolutionary history is a major factor of the lens metabolomic composition”

Line 496: delete “that”

Lines 498-499: the clustering of individualsof the same species is not shown in Fig. 5 but instead in Fig. 3.

Line 560: of the birds taxonomy = of bird phylogeny

Response: Thank you very much! All corrections except two were amended to the text.

Line 365: How do you judge that “The quality of the tree is rather high”?

Response: The description of quality assessment provided earlier: The quality of the obtained trees was monitored according to the following criteria: samples of one species should be in one cluster and do not mix with other species; the Passerides and the Corvides samples should form larger clusters and link with each other to form the Passeriformes cluster; P. cristatus and C. livia should be distant from other clusters.

Fig. 4b: change “higher landbirds” to “Afroaves”

Response: In original paper of Kuhl et al. 2021, this clade is named “higher landbirds”. 

Round 2

Reviewer 1 Report

It was very nice to read evaluate the paper after the review made by authors. Furthermore, I’m gad they found my comments/suggestions useful in improve their work.

I’m satisfied with the sensile improvements and think only a minor formal adjustment are needed. Nevertheless, such adjustments can be perfectly made during a normal, editorial copyediting.

I congratulate authors by their efforts.